# Changes in Land Cover and Urban Sprawl in Ireland From a Comparative Perspective Over 1990–2012

**Achim Ahrens [1],\* and Seán Lyons [1,2]** 

[1]  The Economic and Social Research Institute, Dublin D02 K138, Ireland; Sean.Lyons@esri.ie
[2]  Department of Economics, Trinity College Dublin, Dublin D02 PN40, Ireland
\*   Correspondence: Achim.Ahrens@esri.ie; Tel.: +353-1-8632023

**Abstract:** In this article, we first summarise trends of land use changes and urbanisation in Ireland since 1990 using data from the Corine Land Cover program. In doing so, we compare the developments in Ireland with other European countries. Second, we propose a statistical test for the presence of sprawl using conditional and unconditional convergence tests. The two-part empirical analysis allows us to establish that Ireland has experienced a substantial loss of non-urban land in recent decades. Furthermore, a significant share of urban land use has been extended to remote areas, thereby exacerbating sprawl.

**Keywords:** urban sprawl; land use; land cover change; urban expansion

---

## 1. Introduction

An extensive body of literature shows that spatial development patterns are not necessarily efficient from an economic and environmental perspective and that market failures are pervasive [1,2]. While this provides a key rationale for effective planning practices, these may, in practice, not always succeed in overcoming market failures [3–5]. In the case of Ireland[1], strategic planning practices have been absent or poorly implemented in the crucial Celtic Tiger period of the 1990s and 2000s, when Ireland experienced rapid economic growth and a building boom [6]. Even today, the task of developing sustainable planning policies remains difficult in Ireland, which faces the challenge of addressing severe regional economic imbalances and a shortage of affordable housing. Yet, efficient policies to address market failures can only be designed with a better understanding of past developments.

Economic development is associated with changes in land use, typically transformations from natural green spaces to artificial areas. The expansion of urban space is a common trend across Europe, despite low or stagnant population growth. Indeed, as shown in Figure 1, the growth rate of urban expansion, as measured by the area classified as urban, exceeds population growth in almost all European countries between 2000 and 2012. The trend towards urbanisation of land is especially pronounced in Ireland. With an annual growth rate of 3.1% (2.5%) over 1990–2012 (2000–2012), urban land expansion in Ireland is among the highest in Europe. For comparison, the average rate in other European countries is 1.4% (1.1%) over the same period.

Urban expansion may take place in a dense form, in which new urban areas are located near existing urban structures. The other extreme is that urbanisation can be accompanied by sprawl. Sprawl is a multi-dimensional, ambiguous concept, which can refer to both a state and a process [7]. Due to the complexity of sprawl, there is a large and ongoing literature devoted to developing methods

---

[1]  Throughout this article, Ireland refers to the Republic of Ireland.

to quantify sprawl [8–15]. While substantial advances have been made in the measurement of sprawl, there is a widespread view that no sprawl index on its own can fully represent the complexity of sprawl.

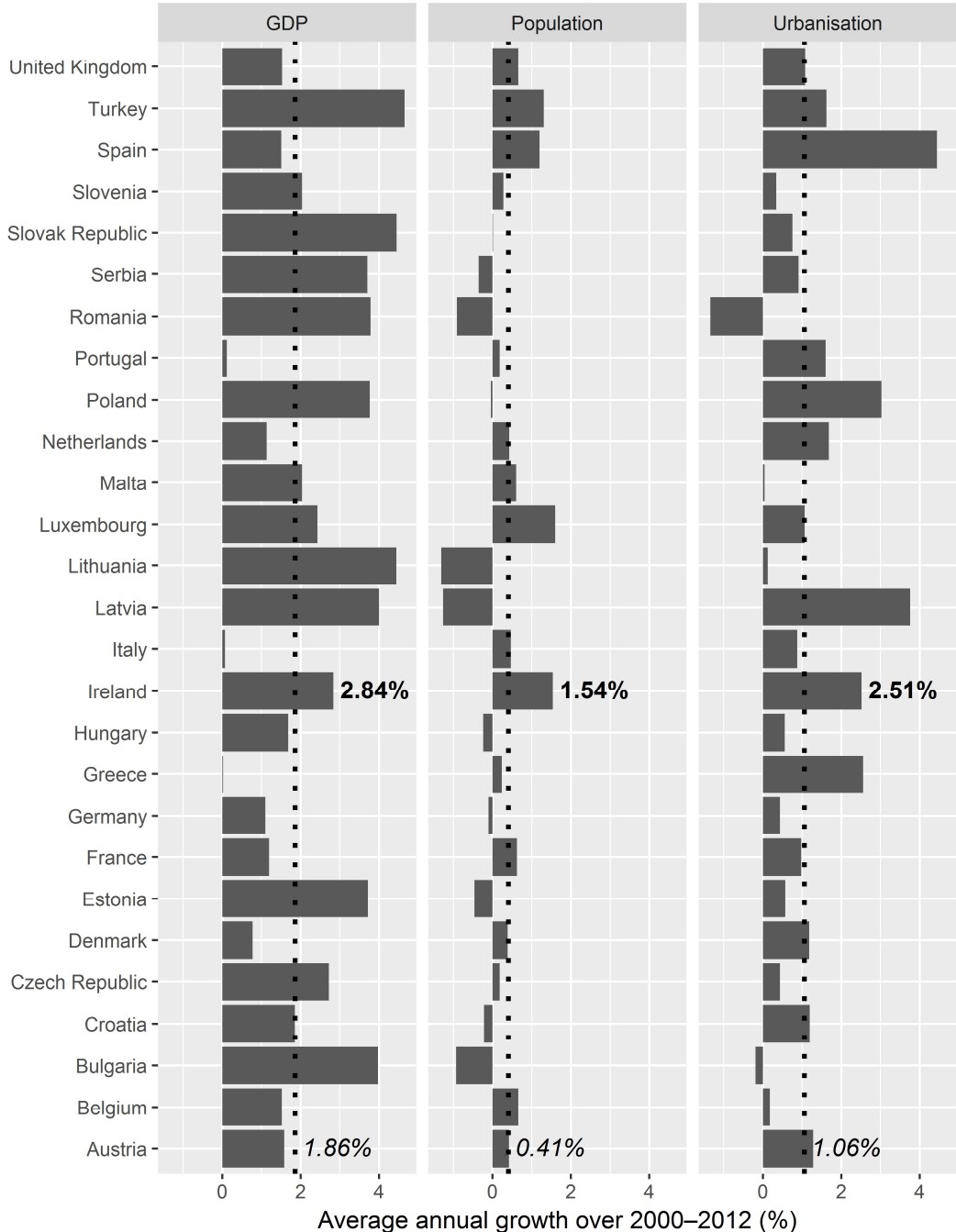

**Figure 1.** GDP growth, population growth and rate of urbanisation in 2000–2012. Dotted lines indicate average annual growth for all countries, excluding Ireland (average rates shown in italics). Growth rates in bold are for Ireland. (*Data source:* Authors' calculations using IMF, Corine Land Cover.).

Sprawl and the consumption of green spaces can have various adverse effects, including negative impacts on health, transport, energy use, the costs of public service provision and the environment [2,16–18]. For example, measures of sprawl are shown to be associated with commuting times and distances driven [19–21]. The increased use of vehicles for commuting and transport has natural implications for greenhouse-gas emissions and pollution [22,23]. Ewing and Rong [24] point out that, because residents in

rural areas are more likely to live in detached single-family houses, sprawl results in higher residential energy use, which again has consequences for the environment. Glaeser and Kahn [25] link sprawl directly to green-house gas emissions and conclude that, using US household data, emissions could be reduced if more people move to cities and locations with a high population density; see also [26–28].

In addition to the literature on the impacts of sprawl, another strand of research investigates what causes sprawl. Population and income growth, supply of undeveloped land and car-based urban structures are often named as determinants of sprawl [1,25,29,30]. Burchfield et al. [30] show that sprawl is more severe in cities with a lack of public transport and with availability of land not subject to planning regulations within the municipality. In a panel study of 282 European cities covering 1990, 2000 and 2006, Oueslati, Alvanides, and Garrod [31] find that the extent of artificial area is increasing in population and income per capita, whereas agricultural productivity at urban fringes alleviates sprawl. Garcia-López [32] find a direct causal effect of highway construction on residential land development when instrumenting highway expansion with the locations of historical road and railroad networks.

For the purpose of this article, we conceptualise the state of sprawl as a land-intensive, low-density and scattered use of land. Similarly, the process of sprawl refers to the process of land use changes towards a higher degree of dispersion, typically accompanied by the consumption of green spaces and creation of artificial areas. To arrive at a higher degree of dispersion, areas with a low level of population density need to exhibit higher growth rates than high-density areas. We will, based on this simple insight, characterise the process of sprawl formally and propose a statistical test for the presence of sprawl analogous to convergence tests from the economic growth literature. Our formulation tests whether land use is moving to a more even, low-density form of land use. While there is a rich literature on the measurement of sprawl, we are not aware of a statistical test in an econometric panel data framework.

Despite the large literature devoted to sprawl, its causes and its consequences, there is no recent analysis examining trends in land use and sprawl for Ireland. This is despite the substantial changes in the use of land Ireland has experienced since the start of the Celtic Tiger period. Understanding trends of urbanisation is especially important since, empirically, urbanisation is only rarely reversed, presumably due to the absence of economic incentives. In this article, we summarise changes in land use for Ireland while comparing these trends to those exhibited by other European countries.

The paper makes contributions in both the research and policy domains. It introduces a formal test for sprawl that requires only readily available data. The paper is relevant to policy in at least two ways. Given the ongoing challenges of improving spatial planning and implementation of regional policy in Ireland, a deeper understanding of past developments is required. Second, Ireland's experiences help demonstrate the possible consequences of failing to implement appropriate spatial planning policies in a period of rapid economic development and construction activity.

This article proceeds as follows. Section 2 presents the methodological framework employed. Sections 2.1 and 2.2 discuss the approach for summarising trends in land cover change in Ireland using Corine Land Cover maps. Section 2.3 proposes a test for sprawl. Section 3 applies the framework to Ireland, and Section 4 provides a discussion of the results and conclusion.

## 2. Methods

### 2.1. Corine Land Cover Data and Methodology

The first part of the empirical analysis focuses on national land cover changes in Ireland based on data from the Corine Land Cover (CLC) programme, which is administered by the European Environment Agency (EEA). Based on satellite images, CLC examines the shape, size, colour, texture and pattern of landscape area and classifies each area of land by type of cover. In the analysis, we compare developments in Ireland on the national level with other European countries.

The Corine nomenclature distinguishes between 44 classes of land cover over three aggregation levels. Table 1 lists the first and second level of the Corine nomenclature. The primary purpose

of Corine is to classify land by land cover types rather than by land use. Land cover refers to the physical characteristics of land, while land use refers to the purpose of land for humans. However, the CLC classification also provides some insights into the use of land. For example, the distinction between artificial and agricultural land indicates whether land is predominantly used for residences and for commercial and industrial activities, or whether the primary purpose is the production of agricultural goods.

**Table 1.** Corine Land Cover classification (level 1 and level 2).

| CLC (level 1) | CLC (level 2) |
| --- | --- |
| 1 Artificial surfaces | 1.1 Urban fabric<br>1.2 Industrial, commercial and transport units<br>1.3 Mine, dump and construction sites<br>1.4 Artificial, non-agricultural vegetated areas |
| 2 Agricultural areas | 2.1 Arable land<br>2.2 Permanent crops<br>2.3 Pastures<br>2.4 Heterogeneous agricultural areas |
| 3 Forests & semi-natural areas | 3.1 Forest<br>3.2 Scrub and/or herbaceous vegetation associations<br>3.3 Open spaces with little or no vegetation |
| 4 Wetlands | 4.1 Inland wetlands<br>4.2 Coastal wetlands |
| 5 Water bodies | 5.1 Continental waters<br>5.2 Marine waters |

We consider the second level of aggregation for our analysis. The 15 second-level categories in Table 1 imply 210 distinct change categories (e.g., 1.1 to 1.2, and 1.1 to 1.3), which we refer to as land cover flows. To summarise land cover flow trends and gain insights into development patterns, we adopt the framework of Feranec et al. [33], who classify land cover flows (LCF) into seven categories. These categories are as follows:

- *Urbanisation* (LCF1) refers to transformation of land, predominantly but not exclusively, agricultural land and forests, into artificial surfaces. These artificial surfaces include areas destined for buildings, industrial facilities and infrastructure, but also artificial green spaces (e.g., parks);
- *Intensification of agriculture* (LCF2) refers to the transition of land from low-intensity agricultural use (pastures [2.3] and heterogeneous agricultural areas [2.4]) to high intensity use (arable land [2.1] and permanent crops [2.2]). High intensity use is thereby understood as being associated with a higher use of artificial fertilisers, weed killers, fungicides and pesticides, as well as the use of modern machinery and techniques, such as irrigation and drainage;
- *Extensification of agriculture* (LCF3) is the converse of intensification (LCF2). Hence, LCF3 refers to the transition from high to low-intensity forms of agriculture;
- *Afforestation* (LCF4) is the re-creation of forest land either naturally or by planting;
- *Deforestation* (LCF5) is the transition of forest land into non-forest land;
- *Water bodies construction and management* (LCF6) refers to the creation of water bodies;
- *Other* (LCF7) represents other non-classified land change flows.

Since we focus on urbanisation, we add another category, de-urbanisation (LCF8), which is the regeneration of artificial areas (i.e., 1.1 to 1.4) into agricultural, semi-natural or natural areas (i.e., 2.1 to 3.3). The LCF classification is shown in Table 2. We employ the LCF classification to summarise trends in land changes over the three sample periods: 1990–2000, 2000–2006 and 2006–2012.

**Table 2.** Classification of land cover flows (LCF) based on Feranec et al. (2010).

| *From* | *To* 1.1 | 1.2 | 1.3 | 1.4 | 2.1 | 2.2 | 2.3 | 2.4 | 3.1 | 3.2 | 3.3 | 4.1 | 4.2 | 5.1 | 5.2 |
|---|---|---|---|---|---|---|---|---|---|---|---|---|---|---|---|
| 1.1 | 0 | 7 | 7 | 7 | 8 | 8 | 8 | 8 | 8 | 8 | 8 | 7 | 7 | 7 | 7 |
| 1.2 | 7 | 0 | 7 | 7 | 8 | 8 | 8 | 8 | 8 | 8 | 8 | 7 | 7 | 7 | 7 |
| 1.3 | 7 | 7 | 0 | 7 | 8 | 8 | 8 | 8 | 8 | 8 | 8 | 7 | 7 | 6 | 7 |
| 1.4 | 7 | 7 | 7 | 0 | 8 | 8 | 8 | 8 | 8 | 8 | 8 | 7 | 7 | 6 | 7 |
| 2.1 | 1 | 1 | 1 | 1 | 0 | 2 | 3 | 3 | 4 | 4 | 7 | 7 | 7 | 6 | 7 |
| 2.2 | 1 | 1 | 1 | 1 | 3 | 0 | 3 | 3 | 4 | 4 | 7 | 7 | 7 | 6 | 7 |
| 2.3 | 1 | 1 | 1 | 1 | 2 | 2 | 0 | 2 | 4 | 4 | 7 | 7 | 7 | 6 | 7 |
| 2.4 | 1 | 1 | 1 | 1 | 2 | 2 | 3 | 0 | 4 | 4 | 7 | 7 | 7 | 6 | 7 |
| 3.1 | 1 | 1 | 1 | 1 | 5 | 5 | 5 | 5 | 0 | 5 | 5 | 5 | 7 | 6 | 7 |
| 3.2 | 1 | 1 | 1 | 1 | 2 | 2 | 2 | 2 | 4 | 0 | 5 | 7 | 7 | 6 | 7 |
| 3.3 | 1 | 1 | 1 | 1 | 2 | 2 | 2 | 2 | 4 | 4 | 0 | 7 | 7 | 6 | 7 |
| 4.1 | 1 | 1 | 1 | 1 | 2 | 2 | 2 | 2 | 4 | 4 | 7 | 0 | 7 | 6 | 7 |
| 4.2 | 1 | 1 | 1 | 1 | 2 | 2 | 2 | 2 | 4 | 4 | 7 | 7 | 0 | 6 | 7 |
| 5.1 | 1 | 1 | 1 | 1 | 7 | 7 | 7 | 7 | 4 | 4 | 7 | 7 | 7 | 0 | 7 |
| 5.2 | 1 | 1 | 1 | 1 | 7 | 7 | 7 | 7 | 4 | 4 | 7 | 7 | 7 | 7 | 0 |

*LCF codes*: 1–urbanisation, 2–intensification of agriculture, 3–extensification of agriculture, 4–afforestation, 5–deforestation, 6–water bodies construction & management, 7–other, 8–de-urbanisation.

## 2.2. Methodology for Distance-Based Analysis

Not all the formation of urban land is qualitatively the same. New urban areas may be developed adjacent to existing urban areas or in remote locations, yielding a scattered, sparse structure of land use. Dense urban structures tend to be more efficient from a land planning perspective; for example, dense structures are associated with lower travel times and thus minimise the environmental impact [20,26]. To assess how Ireland compares to other European countries in this regard, we have calculated the distances from new artificial areas (i.e., Corine class 1) to existing artificial areas, where each area corresponds to a contiguous polygon in the Corine data set. All other things equal, higher distances to existing areas suggest that urban expansion takes place in an unplanned manner, resulting in disperse urban structures, and inefficient economic and environmental outcomes.

## 2.3. Developing a Test for Sprawl

In the final part of the empirical analysis, we develop a formal test for sprawl. For this purpose, we conceptualise sprawl in terms of population and building density. Intuitively, if populations and buildings are clustered in a few areas, the level of sprawl is, with all other things being equal, low. On the other hand, if populations and buildings are distributed uniformly across space, sprawl is at its maximum potential. Accordingly, sprawl as a process is the transition to a state of higher dispersion across space. Sprawl increases over time if low-density areas exhibit higher growth rates in population and building stock than high-density areas. This occurs when people tend to move to low-density areas and when new buildings are constructed in sub-urban areas.

This insight motivates a test of whether growth rates in population and building density are significantly higher in low-density areas. Formally, we consider the model:

$$\Delta density_{i,t} = \alpha + \beta \times density_{i,t-1} + \epsilon_{i,t} \tag{1}$$

where $i$ indexes the unit of analysis. In our case, we use Electoral Divisions (ED), of which there are 3409 in Ireland. The variable $density_{i,t}$ denotes either population density or building stock of ED $i$ at time $t$ in logarithmic terms and $\Delta$ is the first-difference operator.[2] Thus, the left-hand side is the

---

[2]    We use the time index $t − 1$ to indicate the previous observation. For example, the observation before 2011 is 2006.

(approximate) percentage change in density between time $t - 1$ and time $t$. $density_{i,t-1}$ is the density in the previous period and $\alpha$ is the intercept. If the parameter $\beta$ is negative, then areas with a low density in the previous period $t - 1$ tend to exhibit higher growth rates in density between $t - 1$ and $t$. Thus, a negative $\beta$ provides evidence for sprawl. On the other hand, a positive $\beta$ would suggest that, generally, high-density areas exhibit above-average growth, resulting in concentration.

The theoretical framework borrows from the convergence literature in economics, which examines to what extent low-income countries catch up with developed countries [34]. In the convergence literature, a negative $\beta$ is referred to as absolute or unconditional $\beta$-convergence. The model (i) can be extended to include control variables,

$$\Delta density_{i,t} = \alpha_i + \beta \times density_{i,t-1} + x'_{it-1}\delta + \epsilon_{i,t} \tag{2}$$

where $x'_{it-1}$ is a vector of control variables. The parameter $\alpha_i$ represents the ED-level fixed effects which control for unobservable time-invariant and area-specific characteristics determining the attractiveness of EDs. For example, fixed effects may capture that some areas exhibit a higher growth in density due to their proximity to city centres and/or green spaces. If $\beta$ is negative in the above model, we say that conditional $\beta$-convergence is taking place. To provide some intuition for the distinction between unconditional and conditional convergence, suppose that $\beta$ is zero (or positive) in Equation (1), but negative in Equation (2). This would imply that sprawl is only taking place when other factors that determine the attractiveness of residential areas are held constant.

In our analysis, we include the control variables that are relevant for capturing the attractiveness of EDs for residents. Commuting times and distance to motorways[3] measure the proximity to places of work. All other things being equal, we expect EDs with good accessibility to exhibit higher growth rates. Unemployment proxies the lack of employment opportunities and, thus, is likely to be positively associated with population growth. Lastly, we also include broadband usage, which is a relevant location factor in a time of broadband expansion, and the percentage of foreign-born residents as an additional demographic factor. Summary statistics are reported in Table 3.

**Table 3.** Summary statistics.

|  | Obs. | Mean | Std. Dev. | Min | Max |
|---|---|---|---|---|---|
| Population | 10,214 | 1324.6 | 2171.1 | 66 | 38,894 |
| Buildings | 10,323 | 480.9 | 694.6 | 5 | 8615 |
| Commuting (minutes) | 10,214 | 26.1 | 4.4 | 11.68 | 44.5 |
| Unemployment rate (%) | 10,214 | 11.6 | 7.1 | 0 | 56.3 |
| Drive time to motorway (minutes) | 6802 | 47.5 | 40.4 | 1.03 | 212.6 |
| Broadband use (%) | 10,214 | 43.1 | 25.9 | 0 | 95.9 |
| Foreign residents (%) | 10,214 | 2.6 | 3.1 | 0 | 41.5 |

## 3. Results and Discussion

### 3.1. Land Cover Trends in Ireland

In this section, we discuss trends in land cover changes for Ireland based on the LCF classification in Table 2. Figures 2 and 3 and Table 4 were compiled using data from the CLC programme. Figure 2a shows land transformations by LCF class over the period 1990 to 2012 as a share of total land area, while Figure 2b only displays urban-related transformations.

The bar diagrams in Figure 2 show that Ireland experienced an overall higher degree of land conversion relative to other European countries across all LCF classes except LCF 6 (water body construction) and LCF 8 (de-urbanisation). In total, 11.9% of land was transformed between 1990 and

---

[3]　We thank Edgar Morgenroth for providing the motorway distance estimates to us.

2012, compared to 4.2% in Other Europe[4]. The gap between Ireland and Other Europe is especially pronounced for afforestation, intensification and extensification. For example, the proportion of the total land area converted to forests is more than three times as high in Ireland compared to Other Europe (3.9% versus 1.2%). With 32.6% of transformed area, afforestation accounts for the largest share of land transformations in Ireland over the 1990–2012 period, exceeding deforestation (17.1%). For comparison, the rest of Europe experienced a small reduction in forest area.

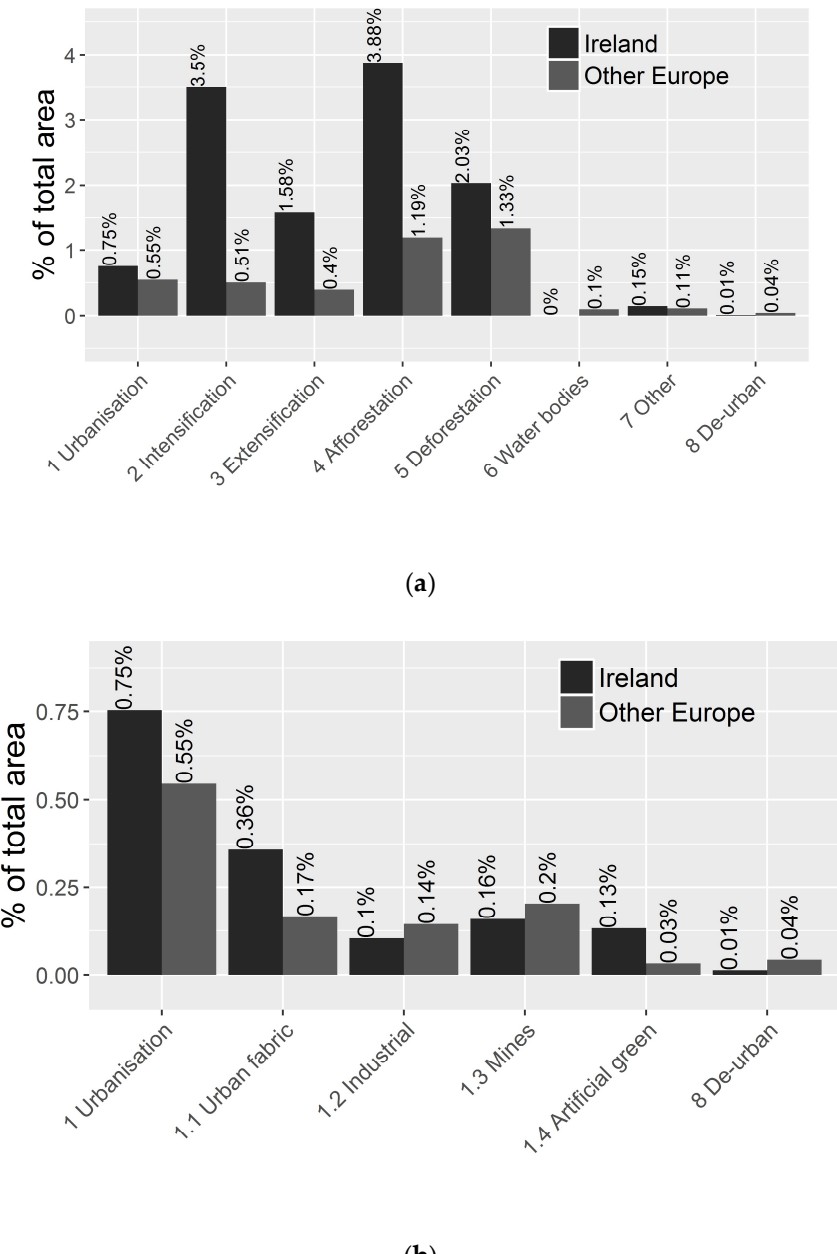

(**a**)

(**b**)

**Figure 2.** Land transformation in Ireland and Other Europe over 1990 to 2012: (**a**) land cover flows in per cent of total area; (**b**) generation of artificial land in per cent of total area.

4    In this section, Other Europe includes the following countries (using iso3 codes): AUT, BEL, BGR, CZE, DEU, DNK, ESP, EST, FRA, GBR, GRC, HRV, HUN, IRL, ITA, LTU, LVA, NLD, POL, PRT, ROU, SRB, SVK, SVN, TUR.

---

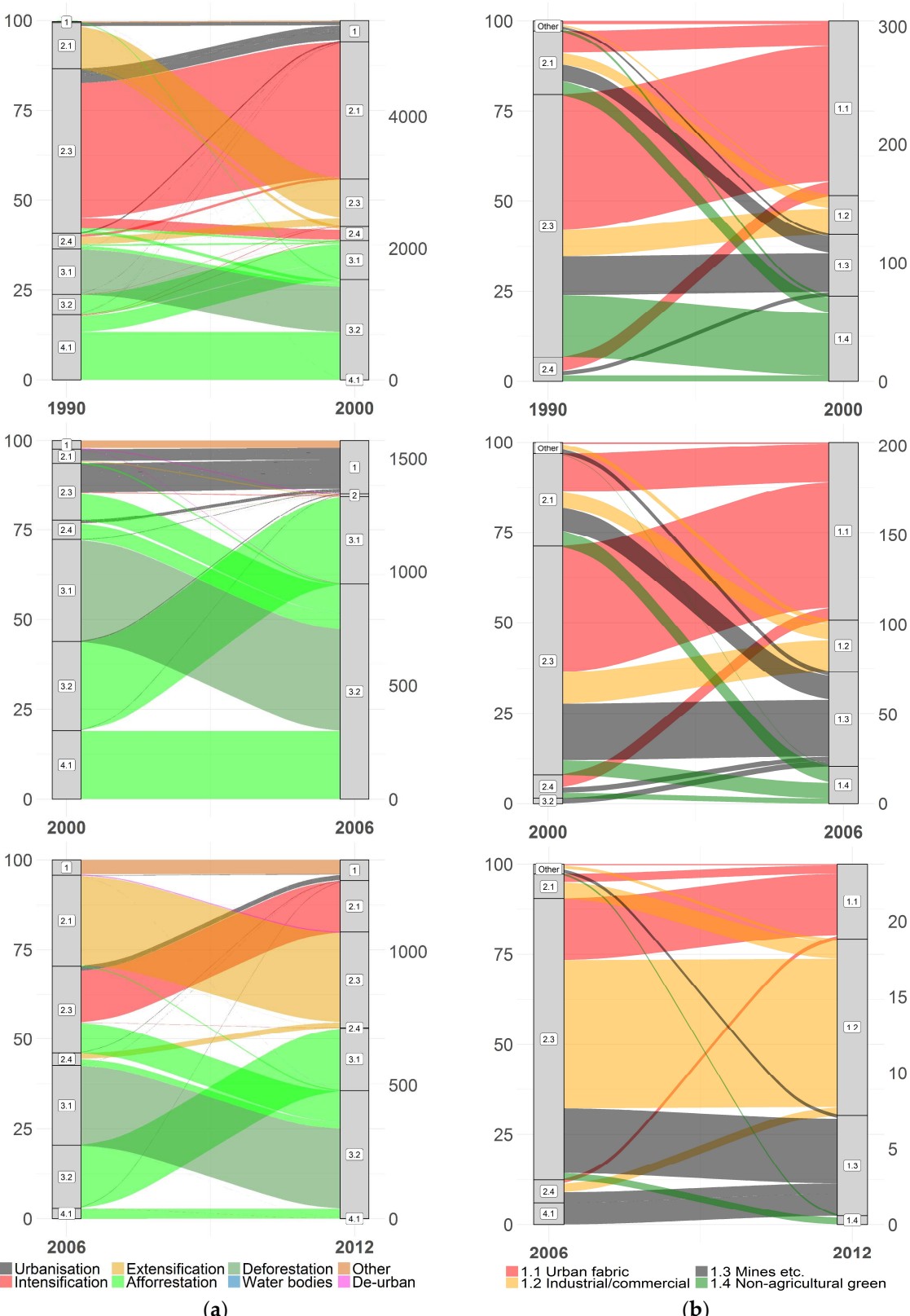

**Figure 3.** Land use flows in Ireland over 1990–2012. Left scale is in per cent of total land cover flows and right scale is in 000 km². Sub-figure (**a**) shows all land cover flows (LCF); sub-figure (**b**) displays land changes leading to the creation of artificial areas.

**Table 4.** Land conversion in Ireland and other European countries over 1990–2012.

| | *Ireland* | | | | *Other Europe* | | | |
|---|---|---|---|---|---|---|---|---|
| | *1990–2000* | *2000–2006* | *2006–2012* | *1990–2012* | *1990–2000* | *2000–2006* | *2006–2012* | *1990–2012* |
| *Share of transformed (%)* | | | | | | | | |
| Urbanisation (LCF 1) | 5.58 | 12.79 | 1.77 | 6.32 | 12.43 | 14.30 | 12.62 | 12.91 |
| *of which* | | | | | | | | |
| Urban (1.1) | 2.70 | 6.30 | 0.37 | 3.01 | 5.31 | 3.30 | 1.69 | 3.93 |
| Industrial (1.2) | 0.60 | 1.82 | 0.86 | 0.87 | 3.25 | 3.48 | 3.66 | 3.41 |
| Mines, etc. (1.3) | 0.95 | 3.36 | 0.49 | 1.33 | 3.01 | 6.68 | 6.75 | 4.80 |
| Artificial green (1.4) | 1.32 | 1.30 | 0.04 | 1.11 | 0.86 | 0.84 | 0.52 | 0.77 |
| Intensification (LCF 2) | 41.50 | 0.25 | 14.45 | 29.39 | 14.03 | 10.00 | 9.56 | 11.97 |
| Extensification (LCF 3) | 13.80 | 0.25 | 26.44 | 13.27 | 12.86 | 3.60 | 7.43 | 9.36 |
| Afforestation (LCF 4) | 26.20 | 56.06 | 30.90 | 32.58 | 31.65 | 22.50 | 26.58 | 28.26 |
| Deforestation (LCF 5) | 12.57 | 28.28 | 22.16 | 17.07 | 23.97 | 43.17 | 36.60 | 31.58 |
| Water bodies (LCF 6) | 0.03 | 0.00 | 0.00 | 0.02 | 2.54 | 1.80 | 2.22 | 2.29 |
| Other (LCF 7) | 0.32 | 2.03 | 3.98 | 1.23 | 1.59 | 3.42 | 3.99 | 2.62 |
| De-urbanisation (LCF 8) | 0.00 | 0.34 | 0.30 | 0.11 | 0.93 | 1.21 | 1.01 | 1.01 |
| *Transformed* | | | | | | | | |
| Area (km$^2$) | 5459.60 | 1579.48 | 1343.18 | 8382.25 | 95,532.21 | 42,382.14 | 46,939.74 | 184,854.09 |
| Area/year | 545.96 | 263.25 | 223.86 | 381.01 | 9553.22 | 7063.69 | 7823.29 | 8402.46 |
| Share of total (%) | 7.76 | 2.24 | 1.91 | 11.91 | 2.18 | 0.97 | 1.07 | 4.23 |
| *Urbanisation* | | | | | | | | |
| Area (km$^2$) | 304.44 | 201.97 | 23.75 | 530.17 | 11,874.89 | 6059.65 | 5922.39 | 23,856.93 |
| Area/year | 30.44 | 33.66 | 3.96 | 24.10 | 1187.49 | 1009.94 | 987.07 | 1084.41 |
| Share of total (%) | 0.43 | 0.29 | 0.03 | 0.75 | 0.27 | 0.14 | 0.14 | 0.55 |
| *Total area (km$^2$)* | 70,366.7 | | | | 4,373,315.7 | | | |

Table 4 provides a detailed break-down by period for Ireland and Other Europe. These data show that afforestation slowed down over the three periods from 143.1 km$^2$ in 1990–2000 to 69.2 km$^2$ per year in 2006–2012, but exceeded deforestation in each period. In relative terms, forest-related flows were especially pronounced in 2000–2006, when afforestation and deforestation jointly accounted for more than 80% of all transformations.

The flow diagrams in Figure 3 illustrate origin-destination flows across all LCF categories (left panel, referred to as 3a) and for urbanisation (right panel, referred to as 3b) split by sub-period for Ireland. These diagrams reveal that afforestation was partially due to the conversion of wetlands (4.1) into forests. Intensification of agriculture is the second largest LCF class, with most of the intensification taking place in the 1990s. Figure 3a shows that in that period, more than 2000 km$^2$ of pastures (2.3) was transformed into arable land (2.1), see top-left diagram. In 2006–2012, 358.6 km$^2$ of new pastures was formed, which, however, did not compensate for the loss in pastures over 1990–2000.

Compared to forest-related and agricultural transformations, urbanisation (LCF1) makes up only a small share of total land transformations, accounting for 6.3% and 12.9% of total transformations in Ireland and Other Europe, respectively. The speed of urbanisation slowed down from 30.4 km$^2$ and 33.5 km$^2$ per year in 1990–2000 and 2000–2006 to only 4.0 km$^2$ over 2006–2012. For comparison, the annualised rate remained more stable in Other Europe, where it decreased from 1187.5 km$^2$ in 1990–2000 to 987.1 km$^2$ in 2006–2012. While the slowdown was more noticeable in Ireland, the impact of urban expansion was generally more pronounced compared to Other Europe, where only 0.55% of the total was urbanised (see Figure 2).

Table 4 also reveals that de-urbanisation occurs only rarely. Only 0.1% of total land transformations in Ireland and 1.0% in the rest of Europe are classified as de-urbanisation (LCF8). This insight stresses the importance of tracking the process of urbanisation due to its lasting impact on the environment.

On the formation side, the creation of urban fabric (1.1) decreased as a share of total urbanisation. While transformation into urban fabric accounted for almost 50% of transitions in land use over 1990–2000, the share dropped to 25% over 2006–2012. Similarly, the formation of non-agricultural green spaces fell from 25% to less than 5%. On the other hand, the share of new land area for industrial purposes increased from less than 10 to 50%. On the consumption side, pastures experienced the largest losses due to urbanisation across all three sub-periods, followed by arable land, whereas only a negligible share of urbanisation involved forests and wetlands.

### 3.2. Sparseness of New Artificial Areas

Summary statistics for the distance-based analysis are shown in Table 5 for Ireland and Other Europe[5]. The proportion of new artificial areas created between 1990 and 2012 that were adjacent to existing structures is slightly lower in Ireland compared to other European countries, with 63.5% versus 64.6%. On the extreme end, 13.3% of new artificial areas were created at least 2 km from existing artificial structures, while the same proportion for Other Europe is only 7.8%. The average distance amounts to 761.1 m, which is considerably higher than in Other Europe (516.9 m). The median distance of nonadjacent artificial areas is 1172.1 m, implying that more than half of new non-adjacent artificial structures were created more than 1.1 km away from existing artificial areas.

**Table 5.** Distance of new artificial areas to existing artificial structures for Ireland and Other Europe for 1990–2012.

| | *Period* | *All Count* | *Adjacent Count* | *in %* | *Distance above (in %)* | | | *Average All* | *Median Non-Adjacent* |
|---|---|---|---|---|---|---|---|---|---|
| | | | | | *20 m* | *200 m* | *2 km* | | |
| Ireland | 1990–2000 | 1137 | 686 | 60.33 | 39.40 | 34.12 | 14.86 | 837.99 | 1146.38 |
| | 2000–2006 | 1564 | 944 | 60.36 | 38.43 | 33.12 | 15.03 | 845.97 | 1359.54 |
| | 2006–2012 | 473 | 384 | 81.18 | 17.97 | 16.28 | 3.81 | 295.39 | 702.71 |
| | Total | 3174 | 2014 | 63.45 | 35.73 | 30.97 | 13.30 | 761.06 | 1172.13 |
| Other Europe | 1990–2000 | 43,431 | 31844 | 73.32 | 25.78 | 22.01 | 5.54 | 389.70 | 775.76 |
| | 2000–2006 | 42,470 | 25688 | 60.49 | 36.71 | 30.97 | 9.15 | 606.50 | 755.59 |
| | 2006–2012 | 43,263 | 25904 | 59.88 | 38.54 | 31.98 | 8.71 | 556.76 | 732.43 |
| | Total | 129,164 | 83436 | 64.60 | 33.65 | 28.30 | 7.79 | 516.94 | 753.47 |

The insight that the process of urban expansion in Ireland is characterised by the formation of scattered, remote urban structures is confirmed by Figure 4, which shows two density curves; one for Ireland (solid line) and one for Other Europe (dashed). The curves are generated using kernel density algorithms and approximate the underlying distribution of distances for Ireland and Other Europe. Compared to Other Europe, Ireland has a lower share of distances below around 1.75 km, while the right tail of the distribution is more pronounced for Ireland, indicating a higher share of artificial formations far away from existing structures.

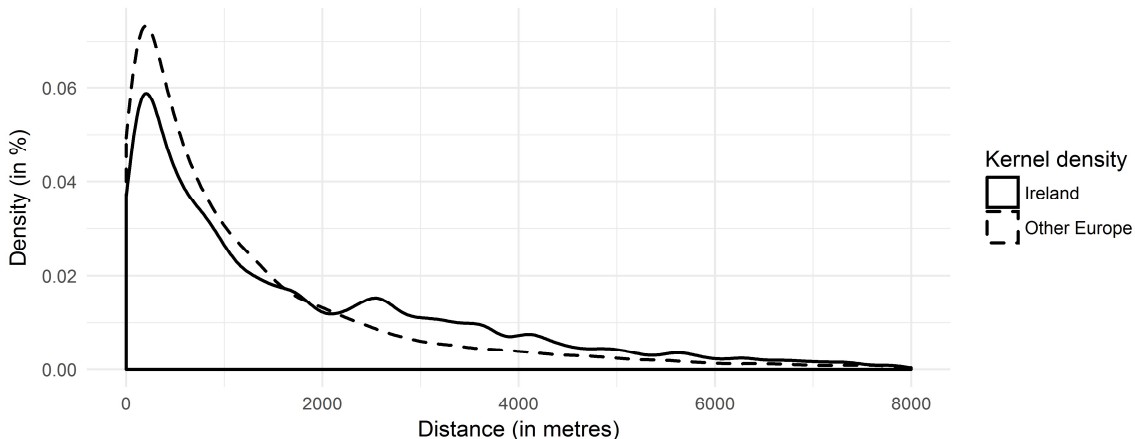

**Figure 4.** Distance of new artificial areas to existing artificial structures for Ireland and Other Europe for 1990–2012.

---

[5]   Due to data constraints, Other Europe uses a reduced sample including AUT, BEL, BGR, CZE, DEU, DNK, ESP, FRA, GRC, HRV, HUN, IRL, ITA, LTU, LUX, LVA, NLD, POL, PRT, ROU, SVK.

Looking at the break-down by period, the average distance in Ireland dropped from above 800 to 295.4 m in 2006–2012, while the share of adjacent new areas rose to 81.2% from around 60.3%. Over 2006–2012, only 3.8% of new artificial areas were located more than 2 km away from existing structures. For comparison, the share of adjacent areas dropped in Europe from 73.3% to only 59.9%. Certainly, as remote locations become increasingly scarce, a fall in average distances is expected; nevertheless, the drop is considerable, especially in comparison to the rest of Europe.

### 3.3. Sprawl Test Results

We estimate different versions of Equations (1) and (2) using a panel dataset of Small Area Population Statistics from the Central Statistical Office covering the Census years 2006, 2011 and 2016. The building stock is taken from the GeoDirectory, which is a registry of buildings in Ireland compiled by the Ordnance Survey Ireland jointly with An Post.

Table 6 shows the estimation results. Columns (i) and (ii) correspond to the basic models in equation (1) with population density and number of buildings per ED, respectively. The $\beta$ coefficient estimates are significantly negative for both population and buildings at the 0.1% level, providing evidence for unconditional $\beta$-convergence across EDs, which, in our context, is evidence for sprawl. However, while highly significant, both coefficient estimates are small in absolute size.

**Table 6.** Testing for sprawl.

|  | (i) | (ii) | (iii) | (iv) | (v) | (vi) |
|---|---|---|---|---|---|---|
|  | *Dependent variable:* **Change in** | | | | | |
|  | **Population** | **Buildings** | **Population** | **Buildings** | **Population** | **Buildings** |
| Lagged level | −0.00277 *** | −0.106 *** | −0.845 *** | −0.855 *** | −0.879 *** | −0.950 *** |
|  | (0.000673) | (0.00390) | (0.0140) | (0.00529) | (0.0171) | (0.00599) |
| $Commute_{i,t-1}$ |  |  | −0.100 *** | 0.00213 | −0.110 *** | 0.00171 |
|  |  |  | (0.0154) | (0.00912) | (0.0154) | (0.00836) |
| $Motorway_{i,t-1}$ |  |  | −0.00674 *** | −0.0123 *** | −0.00291 | −0.00117 |
|  |  |  | (0.00176) | (0.000948) | (0.00191) | (0.000934) |
| $Unemployment_{i,t-1}$ |  |  |  |  | −0.0784 ** | −0.00475 |
|  |  |  |  |  | (0.0257) | (0.0132) |
| $Broadband_{i,t-1}$ |  |  |  |  | 0.0348 *** | 0.0672 *** |
|  |  |  |  |  | (0.00708) | (0.00365) |
| $Foreign_{i,t-1}$ |  |  |  |  | 0.243 | −0.171 *** |
|  |  |  |  |  | (0.187) | (0.0514) |
| Observations | 6805 | 6882 | 6728 | 6728 | 6728 | 6728 |
| Fixed effects | No | No | Yes | Yes | Yes | Yes |

Note: Standard errors in parentheses are robust to within-ED clustering. * $p < 0.05$, ** $p < 0.01$, *** $p < 0.001$. The intercepts in Models (1) and (2) are not shown.

Models (iii) to (iv) add ED-level fixed effects and additional control variables and thus test for conditional $\beta$-convergence. Models (iii) and (iv) include average commuting times in minutes ($Commute_{i,t-1}$) and motorway access ($Motorway_{i,t-1}$) to the model. Motorway access is approximated by the driving time from the EDs centroid to the nearest motorway junction. Both variables are included in logarithmic terms and lagged, i.e., for time $t-1$, to avoid simultaneity bias. When controlling for these transport variables, the $\beta$ coefficients increase in absolute size, suggesting that the process of sprawl is more severe once we control for commuting times and access to motorways. This is in line with theoretical expectations, as it is to be expected that people prefer to migrate to EDs with good motorway access and close to business districts to avoid long commuting times. In other words, low-density areas that are more attractive for commuters are driving sprawl. The coefficients of −0.845 and −0.855 imply that a 1% decrease in density is associated with an increase in the growth rate by approximately 0.85 percentage points. Motorway accessibility only has a negligible association with population density growth, but a stronger association with building construction.

Finally, we consider a more general model in columns (v) and (vi) with unemployment rate ($Unemployment_{i,t-1}$), percentage of non-Irish residents ($Foreign_{i,t-1}$) and percentage of households with

broadband ($Broadband_{i,t-1}$), as a proxy of broadband availability, to the model. The addition of further regressors confirms the presence of conditional $\beta$-convergence, and thus sprawl. Broadband availability is shown to be a strong predictor of both population growth and growth in the housing stock. Areas with high unemployment rates experience lower population growth rates, presumably because they provide fewer employment opportunities, whereas there is no significant effect of unemployment in the previous period on the rate of building construction. The share of foreign-born persons does not appear to be statistically associated with population growth, whereas it is negatively correlated with building growth.

As noted earlier, sprawl is a multi-dimensional phenomenon. Population density and building stock alone do not capture the full complexity of sprawl. However, even when taking these limitations into account, the analysis provides strong statistical evidence supporting the hypothesis that sprawl is occurring in Ireland. Furthermore, the results reveal that population growth rates and the construction of buildings on the level of Electoral Divisions are affected by infrastructure and socio-economic variables. When controlling for these factors, sprawl is shown to be more severe.

## 4. Conclusions

In this article, we show that Ireland has experienced substantial changes in land cover and land use since 1990. Two trends stand out in particular: first, the expansion of areas classified as urban has increased at a higher rate than in the rest of Europe. The process of urbanisation has slowed down in the 2006–2012 period; yet, the changes are likely to be lasting. Since de-urbanisation occurs only rarely, transformations of green spaces to urban areas tend to be permanent. Second, relative to other European countries, urban areas are created in more remote locations, resulting in sprawl. This makes Ireland an important case study for other countries experiencing rapid economic growth and extensive construction activity, but without a spatial framework limiting the unplanned expansion of artificial areas.

Furthermore, we develop and carry out a formal statistical test for sprawl. Using a panel data set of Irish EDs, we find robust evidence that low-density areas exhibit a higher growth in population density and building stock than high-density areas, implying that the population structure is becoming more dispersed over time. The results are more pronounced when controlling for ED-level characteristics. Consistent with theory, population growth and building construction are associated with employment opportunities, broadband availability and motorway access. The proposed sprawl test is generally applicable and relatively straightforward to compute for other jurisdictions. A simple form of the sprawl test only requires demographic and socio-economic data for at least two time periods. The test can be computed for different levels of administrative division, although a sufficient level of disaggregation should be chosen for meaningful insights (e.g., census tracts).

There are some limitations to the analysis presented here, particularly related to the data available for local areas in Ireland. We would have liked to explore associations between sprawl and local variations in income levels or consumer prices, but such data are not available at the Electoral Division level. One potentially interesting extension we hope to pursue in the future is to link census data on commuting flows between small areas to administrative data on local rent levels, thereby directly examining one of the factors encouraging sprawl. People working in areas with relatively high rents should have an incentive to commute rather than live locally, all else being equal. Ireland has experienced rapid rent inflation in recent years, and this seems to be differentiated across the country. This variation could help identify the strength of incentives for moving further from high employment urban areas in response to relatively inflexible housing supply conditions.

Sprawl has been associated with a wide range of adverse effects for the economy and the environment, including increased emissions due to commuting and inefficient residential energy use. To address these adverse effects requires effective policies that can alleviate sprawl. The aim of this study is to summarise land use trends for Ireland and establish the existence of sprawl. Proposing specific policies goes beyond the scope of this work. The existing literature offers multiple pathways for successfully combating sprawl, involving transport policy [35,36], urban growth boundaries [37,38], greenbelts [39] and land taxes [40], among others. The sprawl testing framework is only a first

step towards identifying mechanisms contributing to sprawl and formulating policies. For instance, the significant coefficients on commuting and motorway accessibility suggest that aspects of transport and infrastructure are closely linked to sprawl, which should be explored in a more focused analysis.

The recent Irish National Planning Framework commits to compact development and a further reduction in land use change through the reuse of existing urban areas. It remains to be seen how effective this approach will be in limiting sprawl given the rising demand for housing. The methodology developed in this article can provide a framework for tracking developments in Ireland over time. If the goal of limiting sprawl is achieved, this should be reflected in the scale of the negative coefficient on lagged population in our models. At any rate, future policies will have to be assessed against the background of historic developments, in particular, the uncontrolled urban expansion and consumption of green spaces that took place in Ireland, as demonstrated in this article.

**Author Contributions:** Conceptualization, A.A. and S.L.; formal analysis, A.A.; writing—original draft preparation, A.A.; writing—review and editing, A.A. and S.L.; supervision, S.L.; project administration, S.L.; funding acquisition, S.L.

**Funding:** This research was funded by the ESRI Environment Research Programme, which in turn is funded by the Environmental Protection Agency, Ireland.

**Acknowledgments:** We would like to thank Edgar Morgenroth for helpful discussions and for providing us with motorway distance estimates.

**Conflicts of Interest:** The authors declare no conflict of interest. The funders had no role in the design of the study; in the collection, analyses, or interpretation of data; in the writing of the manuscript, or in the decision to publish the results.

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
