# Peer review of "Changes in Land Cover and Urban Sprawl in Ireland From a Comparative Perspective Over 1990–2012"

_land, doi:10.3390/land8010016_

Round 1
Reviewer 1 Report
The article presents an interesting study. It has potential if some concerns are addressed by the authors. The authors need to rework the paper to address these concerns.
A similar study about European cities including Dublin has been uploaded online as a preprint (Garcia-López, M. À. (2018). All Roads Lead to Rome... and to Sprawl? Evidence from European Cities.). The authors need to reference the study and show how their study is different from the study.
The organization of the paper is not in the normal IMRAD format. That is, Introduction, Methodology, Results and Discussion and Conclusion. The Methodology, Results and Discussion are merged together. The author need to separate them.
The authors need to provide more details about the methods used especially how they implement the models.
The authors need to explain how they analyze the data of the whole Ireland. Did they do it city by city? How many cities were considered? If they did it at the national scale, what is the scale used?
If they did it at the national level, they can choose a couple of cities and do the analysis at a larger scale.
The authors have not provided any map to illustrate the results of their analysis.
Author Response
REVIEWER 1
The article presents an interesting study. It has potential if some concerns are addressed by the authors. The authors need to rework the paper to address these concerns.
We are glad that the reviewer finds our work valuable. We hope that we have been able to address the remaining concerns.
A similar study about European cities including Dublin has been uploaded online as a preprint (Garcia-López, M. À. (2018). All Roads Lead to Rome... and to Sprawl? Evidence from European Cities.). The authors need to reference the study and show how their study is different from the study.
Thanks for pointing us to this interesting work. We have added it to the relevant paragraph, in which we review the literature on the causes of sprawl.
The organization of the paper is not in the normal IMRAD format. That is, Introduction, Methodology, Results and Discussion and Conclusion. The Methodology, Results and Discussion are merged together. The author need to separate them.
We have revised the structure of the draft. Our manuscript includes two separate empirical analyses, which are carried out in Section 2 and 3. Each section is now clearly separated in methodology (2.1, 2.2, and 3.1) and results (2.3, 2.4, and 3.2). If the editor prefers that we should strictly apply the IMRAD format (i.e. consolidating all Methodology in Section 2 and Results in Section 3), we are willing to make the change, even though we think the current structure is most appropriate, given the two-step nature of our work.
The authors need to provide more details about the methods used especially how they implement the models.
We have extended the explanation in Section 3. We also think that the methodology is now more apparent due to clear methodology/results separation, as suggested by the referee.
The authors need to explain how they analyze the data of the whole Ireland. Did they do it city by city? How many cities were considered? If they did it at the national scale, what is the scale used?
The analysis in Section 2 is a country-level analysis comparing Ireland to Other European countries. We have clarified this at the beginning of Section 2. Section 3 uses Electoral Divisions. We hope this is clear in the revised manuscript.
If they did it at the national level, they can choose a couple of cities and do the analysis at a larger scale.
This is certainly a valid approach and would be interesting for future work.
The authors have not provided any map to illustrate the results of their analysis.
The result of our analysis do not come in map form, but we welcome suggestions if the reviewer thinks a specific map would emphasize the results of our work.
Reviewer 2 Report
One major problem of this paper is that it ignores previous studies that quantify cause and effects of land-use change/development to argue that very little paper has done so for sprawl, while this paper itself is using land-use change measure. I think this paper should review some of such papers and indicate clearly what theoretical and methodological progresses they have made based on previous works. For example:
Urban land-use change and environmental market failure:
https://www.mdpi.com/2071-1050/9/1/13/htm
Economic structure and urban land-use change:
https://www.sciencedirect.com/science/article/pii/S0166046217303472
Another issue is that this paper does not adhere to the suggested structure of the journal. Some paper can work in this way, but the structure of this paper is not clear enough and it is better to adhere to the suggested structure.
Line 11: If the negative health and other factors’ connections to sprawl are not tested, they should not appear in abstract.
Line 22: When mentioning literature, the authors should list them out.
Line 64-89: Commuting impacts and cause of sprawl are not studied in later analyses, so I do not think the introduction should devote several paragraphs towards them.
Figure 2: The caption should show what (a) and (b) are. And it is important to tell which year each graph stands for, and whether they are total lands or total transformed lands.
Table 2: The caption of this table also needs improvement, such as to explain what is in% and what is distance above (in%).
Equation 2: The variables in x need to be explained in a variable statistics table.
This paper also misses the most important discussion in sprawl related papers – what are the planning and policy implications? How the results support policy measures that mitigate the negative impacts of sprawl? Such as:
Integrated land and water resource management towards urban sprawl:
https://onlinelibrary.wiley.com/doi/full/10.1002/ldr.3106
Transportation policy to mitigate urban sprawl:
https://www.sciencedirect.com/science/article/abs/pii/S096669231300183X
Growth boundary and urban sprawl:
https://journals.sagepub.com/doi/abs/10.1080/0042098042000214824?casa_token=MVBj8pSDBIkAAAAA%3AptRuNh23ra8z5Hj36j2PRn9saSs8XswTdt7FfY7if-58pF85z2_6wHqNZf142q9WLKK78QMFUhr1VA
Literature of policies should be reviewed and suggested policy path needs to be identified with what the authors find in Ireland.
Author Response
REVIEWER 2
One major problem of this paper is that it ignores previous studies that quantify cause and effects of land-use change/development to argue that very little paper has done so for sprawl, while this paper itself is using land-use change measure. I think this paper should review some of such papers and indicate clearly what theoretical and methodological progresses they have made based on previous works. For example:
Urban land-use change and environmental market failure:
https://www.mdpi.com/2071-1050/9/1/13/htm
Economic structure and urban land-use change:
https://www.sciencedirect.com/science/article/pii/S0166046217303472
It was not our intention to suggest that little work has been done on sprawl, but we acknowledge some of our statements in the introduction were too strong. We have revised the introduction substantially, and included additional literature references for the survey on the measurement, causes and consequences of sprawl. We thank the reviewer for his/her comments.
Another issue is that this paper does not adhere to the suggested structure of the journal. Some paper can work in this way, but the structure of this paper is not clear enough and it is better to adhere to the suggested structure.
We have revised the structure of the draft. Our manuscript includes two separate empirical analyses, which are carried out in Section 2 and 3. Each section is now clearly separated in methodology (2.1, 2.2, and 3.1) and results (2.3, 2.4, and 3.2). If the editor prefers that we should strictly apply the IMRAD format (i.e. consolidating all Methodology in Section 2 and Results in Section 3), we are willing to make the change, even though we think the current structure is most appropriate, given the two-step nature of our work.
Line 11: If the negative health and other factors’ connections to sprawl are not tested, they should not appear in abstract.
This is a valid point. We have modified the abstract accordingly.
Line 22: When mentioning literature, the authors should list them out.
Thanks again for this valid comment. We have corrected this.
Line 64-89: Commuting impacts and cause of sprawl are not studied in later analyses, so I do not think the introduction should devote several paragraphs towards them.
We have shortened the review. However, we think that it is important to provide a brief review of causes and consequences of sprawl in order to demonstrate the relevance of the topic.
Figure 2: The caption should show what (a) and (b) are. And it is important to tell which year each graph stands for, and whether they are total lands or total transformed lands.
We have extended the caption of Figure 2.
Table 2: The caption of this table also needs improvement, such as to explain what is in% and what is distance above (in%).
We have revised the caption of Table 2.
Equation 2: The variables in x need to be explained in a variable statistics table.
We had included a summary statistics table in earlier versions of the manuscript, and excluded the table due to space constraints. Following your suggestions, we have inserted the summary statistics table in the revised manuscript.
This paper also misses the most important discussion in sprawl related papers – what are the planning and policy implications? How the results support policy measures that mitigate the negative impacts of sprawl? Such as:
Integrated land and water resource management towards urban sprawl:
https://onlinelibrary.wiley.com/doi/full/10.1002/ldr.3106
Transportation policy to mitigate urban sprawl:
https://www.sciencedirect.com/science/article/abs/pii/S096669231300183X
Growth boundary and urban sprawl:
https://journals.sagepub.com/doi/abs/10.1080/0042098042000214824?casa_token=MVBj8pSDBIkAAAAA%3AptRuNh23ra8z5Hj36j2PRn9saSs8XswTdt7FfY7if-58pF85z2_6wHqNZf142q9WLKK78QMFUhr1VA
Literature of policies should be reviewed and suggested policy path needs to be identified with what the authors find in Ireland.
Thanks for the suggestion. While it is not the primary purpose of our study to develop or recommend specific policies, we now discuss potential pathways for Ireland.
Reviewer 3 Report
The manuscript analyses changes in land cover in Ireland and tests whether these changes can be associated with urban sprawl. Measuring urban sprawl is a very relevant topic and has been addressed by many scholars and the search for sound, comparable and robust measures is still ongoing. In relation to these efforts, the proposed methodology is rather simplistic and reference to other measures and potential limitations is largely missing. Many previous studies have analysed trends of land cover change using the CORINE dataset. For the empirical test of sprawl the rationale of the chosen fixed effects is not clear and the reader is concerned about causality. Please see my comments for detail. I am concerned about a lack of novelty, interest to an international audience, provision of novel discussion in relation to existing literature and implications for policy-makers or researchers.
Detailed comments
1. Please clarify the contribution of the study and how the new knowledge helps Ireland and the international community
2. The first analysis includes urbanisation and all other land use types unrelated to sprawl. What is the rationale of including LCF2-LCF7 in the analysis and how does it relate to the testing of sprawl? You might want to consider removing these additional classes and focus on an in-depth analysis of LCF1 and LCF8
3. Claims in the introduction are quite strong and I disagree. Please have a closer look at urban economics literature discussing market failures (e.g. Brueckner et al), literature quantifying the effects of spatial development on environmental and economic factors. You will find a huge body of literature which discusses (empirically) various perspectives on ‘effective’ spatial development patterns. This might help you framing a more balanced and informed discussion about urban sprawl.
4. Please explain in more detail the rationale of using the framework by Feranec et al, why you apply this framework and what your contribution is to the methodology. You might want to explain what land cover flows are.
5. Reading section 2.2. left me wondering what type of sprawl you are investigating in section 3 since your analysis shows that the share of industrial land increased most over the analysed period (l. 186). If it is mostly commercial sprawl (buildings), how do you explain the choice of fixed effects?
6. Following this, would it be possible to separate commercial and residential development?
7. Please provide the rational for choosing the fixed effects from the literature. Why broadband connection? How is unemployment rate potentially related to sprawl? Which other fixed effects discussed in the literature did you not include and why not? I recommend extending the analysis and discussion in Section 3 as this might be of most interest to readers
8. I have concerns about causality in the statistical test, even though you mention lagged effects. Please elaborate on this and explain how considering t-1 corresponds to the potential period of lag (same for infrastructure, broad band etc? should it not be different time scales?)
9. Please add a section where you discuss a) limitations of your study, b) your results in relation to other literature c) implications for policy or research, d) interest of the findings beyond Ireland
10. I recommend referring to other literature measuring urban sprawl, how your work relates to these, adds, and potential challenges. Having said this, including an explicit definition of sprawl for your analysis would be beneficial in the introduction.
Author Response
REVIEWER 3
The manuscript analyses changes in land cover in Ireland and tests whether these changes can be associated with urban sprawl. Measuring urban sprawl is a very relevant topic and has been addressed by many scholars and the search for sound, comparable and robust measures is still ongoing. In relation to these efforts, the proposed methodology is rather simplistic and reference to other measures and potential limitations is largely missing. Many previous studies have analysed trends of land cover change using the CORINE dataset. For the empirical test of sprawl the rationale of the chosen fixed effects is not clear and the reader is concerned about causality. Please see my comments for detail. I am concerned about a lack of novelty, interest to an international audience, provision of novel discussion in relation to existing literature and implications for policy-makers or researchers.
Detailed comments
1. Please clarify the contribution of the study and how the new knowledge helps Ireland and the international community
We have revised both introduction and conclusion in order to emphasize the contribution of the manuscript. We acknowledge that the previous manuscript lacked clarity in this respect. In addition to developing a formal test for sprawl, we point out that the case of Ireland provides a relevant case study for other countries, which experience rapid economic growth, but lack an appropriate planning framework.
2. The first analysis includes urbanisation and all other land use types unrelated to sprawl. What is the rationale of including LCF2-LCF7 in the analysis and how does it relate to the testing of sprawl? You might want to consider removing these additional classes and focus on an in-depth analysis of LCF1 and LCF8
The aim of the first part of the analysis is to provide an insight into land cover flows in Ireland. While the focus is certainly on urbanisation, we argue that reporting statistics on LCF2-LCF7 along with LCF 1 & 8 is instrumental in that it provides a context for urban-related land cover flows.
3. Claims in the introduction are quite strong and I disagree. Please have a closer look at urban economics literature discussing market failures (e.g. Brueckner et al), literature quantifying the effects of spatial development on environmental and economic factors. You will find a huge body of literature which discusses (empirically) various perspectives on ‘effective’ spatial development patterns. This might help you framing a more balanced and informed discussion about urban sprawl.
We admit that some of the statements in the introduction were too strong and not sufficiently underpinned by references. We have revised the introduction substantially, and added reference to urban economics and sprawl literature, including the work of Brueckner et al.
4. Please explain in more detail the rationale of using the framework by Feranec et al, why you apply this framework and what your contribution is to the methodology. You might want to explain what land cover flows are.
Thank you for the suggestion. We have pointed out that there are too many distinct change categories (15x(15-1)=210), such that some type of framework is required to summarise land cover flows. We have also defined the term land cover flows.
5. Reading section 2.2. left me wondering what type of sprawl you are investigating in section 3 since your analysis shows that the share of industrial land increased most over the analysed period (l. 186).
Section 3 focuses on the spatial distribution of population and buildings. For instance, a country in which the density of population (or buildings) is equal across space is characterised by a higher degree of sprawl compared to a country where the population (buildings) is concentrated in a few areas. We are testing whether Ireland is moving to a higher degree of sprawl (lower concentration). As we point out, this testing methodology captures only one type of sprawl, but provides relevant insights and can be easily applied to other countries.
If it is mostly commercial sprawl (buildings), how do you explain the choice of fixed effects?
See comment 7 below.
6. Following this, would it be possible to separate commercial and residential development?
Thanks for the suggestion. We agree it would certainly be an interesting research question to distinguish between commercial and residential sprawl. However, we argue this goes beyond the scope of this work. Furthermore, with the data at hand and given the time constraints, we are unfortunately not able to distinguish between commercial & residential. While the GeoDirectory database has information on the primary purpose of buildings (commercial vs residential), we are currently uncertain about this quality of this data, and think this would require further investigation.
7. Please provide the rational for choosing the fixed effects from the literature. Why broadband connection? How is unemployment rate potentially related to sprawl? Which other fixed effects discussed in the literature did you not include and why not? I recommend extending the analysis and discussion in Section 3 as this might be of most interest to readers
We note that the referee seems to use the term fixed effects for explanatory variables more generally. To avoid confusion: we include observed explanatory variables (e.g., unemployment) and, in addition, fixed effects, which control for time-invariant unobserved heterogeneity (e.g., geographic proximity to cities). Fixed effects are equivalent to unit-specific dummies (i.e., least-squares dummy variables) and are conceptually different from observed control variables.
We have clarified the choice of and rational for adding the control variables in Section 3.1. Lastly, we appreciate that the referee thinks that Section 3 is of most interest, and have highlighted the contribution in introduction & conclusion, following the reviewer’s recommendation.
8. I have concerns about causality in the statistical test, even though you mention lagged effects.
We have made no claims about causality, and do no need to. Our parameter of interest is beta, which allows us to test for the presence of sprawl. The interpretation of beta depends on which control variables we include. For example, sprawl may only take place when control variables, such as commuting distance, are held constant.
Please elaborate on this and explain how considering t-1 corresponds to the potential period of lag (same for infrastructure, broad band etc? should it not be different time scales?)
As common in the empirical time-series and panel data literature, t-1 denotes the lag of an observation. For example, y(t-1) is the observation that occurs before y(t). Since our data set has observations for 2006, 2011, and 2016, t-1 refers to the previous census year. We think this is less misleading than using t-4. We have clarified this in footnote 4. If the referee prefers us to use y(t-4), we can change that.
9. Please add a section where you discuss a) limitations of your study, b) your results in relation to other literature c) implications for policy or research, d) interest of the findings beyond Ireland
Thank you for your suggestions. We revised and extended both introduction and conclusion to motivate our study thoroughly, and discuss its implications for other countries and policy making.
10. I recommend referring to other literature measuring urban sprawl, how your work relates to these, adds, and potential challenges. Having said this, including an explicit definition of sprawl for your analysis would be beneficial in the introduction.
Thank you again. We have expanded on the paragraph that defines sprawl.
Round 2
Reviewer 1 Report
The author(s) have revised the paper accordingly except that the structure is still not appropriate.
Now the article does not have a Conclusion section. The Discussion section should be changed to the Conclusion section and the Methodology should be separated from the Results Section. The Results section can be named Results and Discussion. It does not matter if they have two or more different analysis and validations. The methodology section can have multiple subsections.
2.0 Methodology
2.1 CORINE Land Cover data
2.2 Distance-based analysis
2.3 Developing a test for sprawl
3.0 Results and Discussion
3.1 Land-cover trends in Ireland
3.2 Sparseness of new artificial areas
3.3 The results of the models of sprawl detection
4.0 Conclusion
Author Response
We have adjusted the structure of the manuscript following the suggestions of the referee. Metholodogy and Results are separately presented in Sections 2 and 3.
Reviewer 2 Report
The paper has addressed my concerns and can be published. Some checks and corrections can further improve it.
Author Response
Thank you. We have made some further changes to the structure in response to Reviewer 1, and some more minor additions and corrections throughout the paper.
Reviewer 3 Report
Thank you for addressing my review comments in the revised version. The paper has been greatly improved, guides readers well through the analysis and reads well. My main comment at this stage is that you might want to think about discussing more explicitly how your analysis can be useful to other countries and how your framework can be replicated and used by other countries. This is an interesting contribution which I think can be further discussed (more explicitly).
In lines 328 you refer to policy measures discussed in other studies since making explicit policy recommendations is beyond the scope of your work as you say. Nonetheless, I think you could be more specific in what the results of your statistical test might imply for policy (starting points) and discuss your own work more in the last paragraph.
Section 4 would also benefit from a discussion of limitations of the framework, which other data might be interesting to be used and avenues for future research. It what way might Ireland be specific? What needs to be in place for other case studies to apply the framework? Which challenges might you see?
Author Response
(Comments in italics. Responses in bold.)
Thank you for addressing my review comments in the revised version. The paper has been greatly improved, guides readers well through the analysis and reads well. My main comment at this stage is that you might want to think about discussing more explicitly how your analysis can be useful to other countries and how your framework can be replicated and used by other countries. This is an interesting contribution which I think can be further discussed (more explicitly).
Good point: we now emphasise that the method would be relatively straightforward to apply in other jurisdictions and does not impose onerous data requirements.
In lines 328 you refer to policy measures discussed in other studies since making explicit policy recommendations is beyond the scope of your work as you say. Nonetheless, I think you could be more specific in what the results of your statistical test might imply for policy (starting points) and discuss your own work more in the last paragraph.
We now note that the method could be used to in tracking implementation of the National Planning Framework and are more explicit about how successful policy would affect the metric.
Section 4 would also benefit from a discussion of limitations of the framework, which other data might be interesting to be used and avenues for future research. It what way might Ireland be specific? What needs to be in place for other case studies to apply the framework? Which challenges might you see?
We have added a discussion of the limitations, which we see as mainly arising from the data one can obtain at sufficiently disaggregated level. We also set out a potentially fruitful direction for future work, moving towards unpacking the factors driving sprawl.